# Development and Evaluation of Non-Antibiotic Growth Promoters for Food Animals

**DOI:** 10.3390/vetsci11120672

**Published:** 2024-12-21

**Authors:** Hanfei Wang, Hengji Zhao, Bocheng Tai, Simeng Wang, Awais Ihsan, Haihong Hao, Guyue Cheng, Yanfei Tao, Xu Wang

**Affiliations:** 1National Reference Laboratory of Veterinary Drug Residues (HZAU), Veterinary Medicine Research Center, Huazhong Agricultural University, Wuhan 430070, China; wanghf@webmail.hzau.edu.cn (H.W.); zhaohengji@webmail.hzau.edu.cn (H.Z.); bocheng99@webmail.hzau.edu.cn (B.T.); wsmsimeng@163.com (S.W.); haohaihong@mail.hzau.edu.cn (H.H.); chengguyue@mail.hzau.edu.cn (G.C.); 2Department of Biosciences, COMSATS University Islamabad, Sahiwal 44000, Pakistan; awais@cuisahiwal.edu.pk

**Keywords:** astragalus, hawthorn, broiler, growth performance, gut microbiota

## Abstract

Antibiotic growth promoters are used now but cause antibiotic resistance. Our aim is to find non-antibiotic ways to make food animals grow faster. We tested plant extracts in mice and broiler chickens. A combination of hawthorn and astragalus extracts made mice grow better by promoting muscle growth and immunity. A further extract improved growth, feed conversion, digestive enzymes, and gut microbiota in mice and broilers. Adding a certain amount of extract to the broilers’ drinking water increased muscle content and carcass quality. These findings offer new ways to stop antibiotic misuse and help develop better growth promoters for food animals.

## 1. Introduction

Livestock and poultry growth is closely associated with people’s standard of living. The faster the growth rate of livestock and poultry, the lower the price of meat, eggs, and dairy products in people’s lives and the higher their standard of living. Therefore, antibiotic growth promoters (AGPs) have been extensively utilized in livestock production worldwide in recent decades to enhance the growth rate of livestock and poultry by maintaining animal health and improving feed efficiency [1,2]. AGPs inhibit pathogens and, at the same time, harm the gut microbiota, reducing competition for nutrients in feed from the gut microbiota, increasing the amount of nutrients accessible to food animals and boosting their growth rates. However, the misuse of AGPs has given rise to the emergence of pathogens resistant to fluoroquinolones, vancomycin, and third- and fourth-generation cephalosporins, resulting in a ban on the use of all AGPs in feed in the European Union [3,4]. Shortly thereafter, the US Food and Drug Administration (FDA) prohibited the use of enrofloxacin in poultry due to the increase in fluoroquinolone-resistant Campylobacter, a trend that is positively correlated with a rise in the use of the drug in the poultry industry [1]. In China, the improper use of antibiotics in livestock and aquaculture has also led to serious antibiotic residues and environmental pollution. Moreover, antibiotic residues accelerate the development of antibiotic resistance in animals and humans, posing a threat to human health [5]. Consequently, political and consumer pressure has contributed to a decrease in the utilization of AGPs in the production of food animals. Thus, there is an urgent need to develop alternatives to AGPs to fulfill the increasing demand for livestock and poultry while minimizing the risk of resistance selection and its consequences for human and animal health [1,2].

Natural plant extracts are regarded as safer and more environmentally friendly alternatives to AGPs [6]. They do not directly eliminate harmful bacteria in the gut, but instead promote growth through other means, significantly reducing the risk of drug resistance. For instance, adding phenolic extracts of blueberry (*Vaccinium corymbosum*) and blackberry (*Rubus fruticosus*) to the diet can significantly promote growth in chickens [7]. Huangqi (also known as astragalus, the dried root of the legume *Astragalus mongolica* or *Astragalus membranaceus*) and chao shanzha (the stir-fried fruit of *Crataegus pinnatifida*) are two natural plants frequently employed in traditional Chinese medicine. Huangqi has a long utilization history and is considered to be the prime medicine for replenishing ‘chi’ [8]. Modern medical research has indicated that the main constituents of huangqi are astragalus polysaccharides and astragaloside, which possess antioxidant and anti-inflammatory properties [9,10,11]. Shanzha (also known as hawthorn) and its derivatives are widely used as food and complementary medicine worldwide. Heating and thermal processing are commonly reported in the preparation of hawthorn. Heat treatment alters the medicinal application and efficacy of hawthorn. Chao shanzha is a stir-fried shanzha that has been reported to be effective in ameliorating indigestion. Chao shanzha has a high content of glucose-1-phosphate and lignan, which might be associated with an increase in gastrin and a decrease in vasoactive intestinal peptides [12]. However, there are no studies regarding chao shanzha and huangqi as growth promoters. Chaonengsu (CNS) is a commercialized liquid chicken feed additive consisting of various amino acids and vitamins that promotes the growth of chickens and improves their immunity, yet its effect is not remarkable. Guanidinoacetic acid (GAA) is a precursor of creatine, which is widely present in muscle and nerve tissues and is the main energy-supplying substance in animal muscle tissues; it has been extensively used in feed additives as a growth promoter [13]. Consequently, this research aims to develop a growth promoter based on chao shanzha and huangqi extracts. Additionally, we also intend to optimize and enhance the growth-promoting effects of the CNS.

In this study, we first extracted the active ingredients from chao shanzha and huangqi and mixed them to develop a new liquid chicken feed additive, named SQ, which is capable of promoting muscle growth and enhancing immunity in mice. To further optimize the SQ and CNS, we mixed the SQ and CNS in various ratios to produce a composite super energy extract (CSEE). The most effective CSEEs were selected based on their growth-promoting performance, survival rate, digestive enzyme activity, glucose and lipid serum levels, and so on. Finally, the effective concentration of CSEEs was investigated in broiler chickens. This study provides two novel, safe, and environmentally friendly non-antibiotic growth promoters (NAGPs), which can enhance broiler productivity and contribute to further research in this field.

## 2. Materials and Methods

All experimental procedures were conducted in accordance with the Guidelines for Experimental Animals established by the Ministry of Science and Technology of the People’s Republic of China and were approved by the Animal Care Committee of Huazhong Agricultural University.

### 2.1. Materials

Chao shanzha was obtained from Hubei Qiangkang Traditional Chinese Medicine Beverage Co., Ltd., Wuhan, China; Huangqi from Gansu Anyuan Pharmaceutical Co., Ltd., Dingxi, China and Chaonengsu (CNS, components listed in Appendix A) from Foshan Telimax Biotechnology Co., Ltd., Foshan, China. The glucose (Glu), triglyceride (TG), total cholesterol (TC), low-density lipoprotein cholesterol (LDL-C), high-density lipoprotein cholesterol (HDL-C), and total bile acid (TBA) test kit were purchased from Shenzhen Myri Biomedical Electronic Co., Ltd., Shenzhen, China. The pepsin (A080-1-1), alpha amylase (C016-1-1), and lipase (A054-1-1) test kit were purchased from Nanjing Jiancheng Technology Co., Ltd., Nangjing, China. Guanidine acetic acid (GAA) was provided by Ybolay Food Ingredients Co., Ltd., Yangjiang, China.

### 2.2. Cell Culture

The mouse myogenic cell line C2C12 was obtained from the Key Laboratory of Agricultural Animal Genetics, Breeding and Reproduction of Ministry of Education (HZAU), and the mouse macrophage line RAW264.7 was obtained from the National Reference Laboratory of Veterinary Durg Residues (HZAU). C2C12 and RAW 264.7 cells were cultured in Dulbecco’s modified Eagle’s medium (DMEM; Gibco, Waltham, MA, USA) supplemented with 10% fetal bovine serum, 100 U/mL penicillin and 100 µg/mL streptomycin in an incubator at 37 °C and 5% CO_2_.

### 2.3. CCK8 Assay

The experimental protocol was as described in the previous studies [14]. Specifically, the C2C12 cell suspension was inoculated in a 96-well plate and pre-cultured for 12 h. The cells were treated with or without 15.6 mg/mL SQ for 12, 24, or 36 h. An amount of 10 μL of CCK8 solution (BS350B, Biosharp, Hefei, China) was added, and the cells were incubated in a 37 °C and 5% carbon dioxide incubator for 2 h. Then, the optical density (OD) was determined using an absorbance microplate reader at 450 nm. The cell survival rate was calculated as follows: ((OD value of the experimental group − OD value of the blank group)/(OD value of the control group − OD value of the blank group)) × 100%. Among them, the OD value of the experimental group was the OD value containing DMEM medium, C2C12 cells, SQ, and CCK8 solution; the OD value of the control group was the OD value containing DMEM medium, C2C12 cells, and CCK8 solution; the OD value of the blank group was the OD value containing DMEM medium and CCK8 solution.

### 2.4. EdU Assay

C2C12 cells in the logarithmic growth phase were harvested, and a cell climbing slice was placed on the bottom of a 24-well plate. C2C12 cells were inoculated into 24-well plates at 5 × 10^5^ cells/mL for 12 h. Then, the cells were treated with serum-free DMEM solution containing/without 15.6 mg/mL SQ for 36 h. The cells were incubated with 5 μM BeyoClick™ EdU-555 working solution (C0075S, Beyotime, Shanghai, China) for 2 h. The medium was discarded. The cells were washed twice with PBS, and 250 μL of 4% paraformaldehyde was added to the slides of each well. After 15–30 min of incubation at room temperature, the fixative was removed. The cells were washed twice with PBS again, and 500 μL of 3% bovine serum albumin (BSA) solution per well was added for blocking. After the removal of the wash solution, 500 μL of 0.5% Triton X-100 was added to each well for permeabilization and incubated for 20 min at room temperature. Subsequently, 200 μL of click additive solution was added to each well and incubated for 30 min at room temperature in the dark. After the removal of the reaction mixture, each well was washed twice with 500 μL of PBS. Then, 500 μL of 5 μg/mL Hoechst 33342 solution per well was added and incubated for 15–30 min at room temperature in the dark. The wash solution was removed by washing twice with PBS. The cell climbing films in the wells were carefully clamped with clean tweezers, placed on slides and observed under an inverted fluorescence microscope.

### 2.5. Neutral Red Phagocytosis Assay

Logarithmically growing RAW264.7 cells were inoculated into 24-well plates and subjected to drug treatment upon reaching 60% confluence. Control and SQ groups were established. After 24 h of treatment, three replicate wells in each group were treated with neutral red staining solution. Once the drug incubation of the cells was completed, the medium was removed, the cells were washed once with PBS, the washing solution was discarded, 200 μL of Neutral Red Staining Solution (C0125, Beyotime, Shanghai, China) was added to each well for 10 min of staining. After staining, the staining solution was discarded, and the cells were washed twice with PBS for observation and picture were taken.

### 2.6. Hematoxylin and Eosin (H&E) Staining

The mouse muscle H&E staining protocol was performed as described in previous studies [15]. In brief, after euthanizing the mice, the leg muscle tissues were collected, fixed with 4% paraformaldehyde and embedded in paraffin. Subsequently, the muscle tissues were subjected to pathological sectioning and H&E staining. The sections were observed and captured using an upright microscope (Olympus, BX53, Tokyo, Japan). Three random fields of view were selected for each section with a magnification of 40×. The captured images were analyzed and evaluated by professional pathological laboratory technicians. Furthermore, we conducted quantitative analysis of the muscle indicators in the tissues and statistically analyzed the number and diameter of muscle fibers in the muscles.

### 2.7. Real-Time Quantitative Polymerase Chain Reaction (qPCR)

qPCR was used to detect the mRNA expression of myogenic regulatory factors in mouse muscle and inflammatory factors in the spleen. The experimental protocol was performed as described in a previous study [14]. In brief, total RNA from total cells or muscle tissues was extracted using an RNA isolater total RNA extraction reagent (R401-01, Vazyme, Nangjing, China). cDNA was synthesized under a total reaction volume of 20 µL using the HiScript II 1st strand cDNA synthesis kit (R212, Vazyme, China). An amount of 50 ng cDNA and 5 nM primers were used for each RT-qPCR reaction. Pairs were performed using 2× universal SYBR green fast qPCR mix (RK21203, ABclonal, Wuhan, China). The results were monitored using a CFX384 real-time PCR detection system (Bio-Rad, Hercules, CA, USA), which was programmed for one 10-min cycle at 95 °C, followed by 39 cycles of 10 s at 95 °C and 10 s at 60 °C, respectively. Relative expression levels were calculated using the 2^−∆Ct^ method. Mouse-specific primers were designed via the NCBI website (Appendix A) using GAPDH and β-actin as internal reference genes. The primers were synthesized and supplied by Sangon Biotech Co., Ltd. (Shanghai, China).

### 2.8. Preparation of SQ

Samples of each chao shanzha and huangqi were weighed (20 g each), 6× water (*v*/*w*) was added, and the mixture was soaked for 15 min. The mixture was then heated to a boil at 420 °C, kept at 300 °C for 20 min, and filtered through eight layers of sterile gauze to obtain the first decoction. Water (6×) was added to the remaining dregs, after which the mixture was boiled at 420 °C, kept at 300 °C for 15 min, and filtered through eight layers of sterile gauze to obtain the second decoction. The two decoctions were mixed and concentrated to 1 g/mL at 350 °C and stored at 4 °C.

### 2.9. CNS and SQ Intercalation Tests

Chinese herbal plant extracts are generally composed of a variety of substances that can vary widely in water solubility and are therefore relatively unstable, difficult to absorb, and have low application potential [16]. To increase the application potential of SQ, we mixed SQ, consisting of a mixture of chao shanzha and huangqi extracts, with CNS in different proportions. On the one hand, moderate supplementation with vitamins and amino acids can balance the diet and enhance the growth-promoting effect; on the other hand, we hoped that the mixed liquid feed additive would be more stable. CNS and SQ were mixed at ratios of 10:1, 5:1, 1:1, 1:5, 1:10 (*v*/*v*), etc., and the solution stability and color changes were observed.

### 2.10. CSEE Thermal Stability Test

The following test solutions were prepared: SQ (1 g/mL), CNS, CSEE-H (SQ:CNS = 1:5 (*v*/*v*)), CSEE-M (SQ:CNS = 1:8 (*v*/*v*)), and CSEE-L (SQ:CNS = 1:10 (*v*/*v*)). Five hundred microliters of each of the above test solutions were taken, sealed, placed in a thermostatic electric oven at 60 °C, and photographed for observation on days 0, 5, and 10.

### 2.11. Mouse Experiment

Sixty male KM mice (SPF grade, weighing 18–20 g) were provided by Huazhong Agricultural University (license number SCXK (E) 2020-0019). The experiment was approved by the Scientific Ethics Committee of Huazhong Agricultural University (ID number: HZAUMO-2022-0178). Prior to the experiment, the mice were allowed to drink and eat freely for at least 5 days to acclimate to the feeding environment. Sixty mice were divided into six groups of 10 mice each: the blank group, the SQ group (1 g/mL), the CNS group, the CSEE-H group (SQ:CNS = 1:5 (*v*/*v*)), the CSEE-M group (SQ:CNS = 1:8 (*v*/*v*)), and the CSEE-L group (SQ:CNS = 1:10 (*v*/*v*)). The treatment was continued for 14 days. During the feeding period, the mice were weighed every two days for body weight and food weight, and the feed–gain ratio (F/G) and average daily gain (ADG) were calculated. The growth-promoting performance of the SQ, CNS, and CSEE groups (CSEE-H, CSEE-M, and CSEE-L) was evaluated by analyzing the body weight of the mice. After the experiment, the mice were euthanized and the spleen, thymus, and muscle tissues of the left leg of the mice were collected. The spleen or thymus index of mice = ((weight of spleen or thymus)/body weight of mice) × 100%.

### 2.12. Tail Suspension Test

The tail suspension test (TST) is a behavioral test in mice that can be used as a method to screen for the efficacy of potential antidepressants, stimulants, and sedatives, and to evaluate other experiments that may affect depression-related behavior [17]. In the present study, we used the TST to assess the vitality of the mice after drug administration. The main procedure was the same as that used by Adem Can et al. [17]. The following simplifications were used: the same batch of mice was separated by cardboard to prevent them from learning each other’s behavior; the cardboard was cut to a length of 15 cm; the experiment was conducted in an SPF animal house in a quiet environment without the use of white noise equipment; the mice used in this experiment were KM mice, which do not exhibit tail climbing behavior, so no tail-crawling equipment was used; and the behavioral analysis focused on whether the mice moved their forelimbs.

### 2.13. Forced Swimming Experiment

The forced swim test (FST) is also a behavioral test in mice and, like the TST, has been used to assess other behaviors that may influence depression-related behavior [18]. In the present study, we used the FST to assess the vitality of the mice after drug administration. The main procedure was the same as that described by Adem Can et al. [18]. The following simplifications were used: the same batch of mice was separated by cardboard to prevent them from learning each other’s behavior; the height of the tank was 22.5 cm, the diameter was 15.5 cm, and the height of the water surface was 12 cm; the behavioral analysis was concerned with recording only the time of displacement of the mice; floating and movements to maintain body balance were neglected.

### 2.14. Broiler Experiment

A total of 180 one-day-old male Arbor Acres (AA) broiler chicks were purchased from Henan Longhua Herding Co., Ltd., Zhengzhou, China. This experiment was approved by the Scientific Ethics Committee of Huazhong Agricultural University (ID number: HZAUCH-2023-0003). Broiler chickens were randomly divided into 5 groups (A to E), with 3 replicates in each group, and each replicate contained 10 chickens. The blank group (A) received a basal diet (composition shown in Appendix A); the positive control group (B) received a basal diet supplemented with 600 mg/kg GAA [19]; and the experimental groups (C–E) received 0.5%, 1% or 2% CSEE-M solution in the drinking water. The experiment lasted for 42 days and was divided into three phases: days 0–14, days 15–28, and days 29 to 42 [20]. For each replicate, chickens were weighed on an empty stomach before morning feeding on days 14, 28, and 42. The average daily feed intake (ADFI), ADG, and F/G were calculated for the three stages, from 1–14, 15–28, and 29–42 days of age. The birds were observed daily, and mortality was recorded. Before all the broilers were killed, 10 broilers from each group were selected, slaughtered, cleaned, and processed, the right leg was skinned and weighed, and the carcass ratio and leg muscle ratio of the broilers were calculated. Five of the ten slaughtered broilers in each group were randomly selected, and the duodenum was removed for digestive enzyme activity testing.

### 2.15. Gut Microbiota

When slaughtering broilers, five broilers were randomly selected from the blank group, positive control group, and 0.5% CSEE-M group, and fresh intestinal contents were collected and quickly frozen at −80 °C. A 16S sequencing process was carried out at Beijing Novogene Co., Ltd., Beijing, China. The experimental method has been described previously [21,22,23]. Total genomic DNA from the samples was extracted via the CTAB/SDS method, and the concentration and purity were monitored via 1% agarose gels. The processed DNA was diluted to 1 ng/L with sterile water. Phusion High-Fidelity PCR Master Mix (New England Biolabs, Ipswich, USA) was used for PCR analysis. A TruSeq DNA PCR-Free Sample Preparation Kit (Illumina, San Diego, USA) was used to generate sequencing libraries and add index codes. Library quality was assessed via a Qubit 2.0 fluorometer (Thermo Scientific, Waltham, MA, USA) and an Agilent Bioanalyzer 2100 system, and libraries were sequenced via the Illumina HiSeq 2500 platform. The data were rarefied to the average number of reads detected in a single sample. Operational taxonomic units (OTUs) were clustered with a 97% similarity cutoff via UPARSE (version 7.0.1090; http://drive5.com/uparse/, accessed on 15 May 2023). The taxonomy of each 16S rRNA gene sequence was analyzed via the RDP Classifier (version 2.11; http://sourceforge.net/projects/rdp-classifier/, accessed on 15 May 2023) against the Silva (SSU115) 16S rRNA database, with a confidence threshold of 70%. Sequence analysis was performed via USEARCH software (version 10; http://drive5.com/usearch, accessed on 15 May 2023). A denoised sequence is called a ‘ZOTU’ (zero-radius OTU). Each ZOTU was screened for further annotation. For each representative sequence, the Greengenes Database (release 13.5; http://greengenes.secondgenome.com/, accessed on 15 May 2023), which is based on the RDP classifier algorithm, was used to annotate the taxonomic information. OTU abundance information was normalized using a standard sequence number corresponding to the sample with the fewest sequences. Subsequent analyses of alpha diversity and beta diversity were performed on the basis of these output normalized data. The Shannon diversity index and observed OTU levels were used as alpha diversity metrics. All these indices in our samples were calculated via QIIME (version 1.7.0; http://qiime.org/scripts/assign_taxonomy.html, accessed on 15 May 2023) and displayed via R software (version 2.15.3; R Project). Beta diversity analysis was used to evaluate differences in the species complexity of the samples; beta diversity was calculated via unweighted UniFrac and assessed via principal component analysis (PCA) via QIIME (version 1.7.0) with R software (version 3.3.1).

### 2.16. Statistical Analysis

All the statistical analyses in this study, except for the macrogenomic dataset, were performed via GraphPad Prism 8. Analysis by *t*-tests (Student’s *t*-tests) or one-way ANOVA (*t*-tests were used for comparisons between two groups, and one-way ANOVA was used for comparisons among three or more groups). All the results were considered significantly different when the *p* value was less than 0.05. ns: *p* > 0.05; *: *p* < 0.05; **: *p* < 0.01; ***: *p* < 0.001; ****: *p* < 0.0001.

For the macrogenomic dataset, the raw data for each sample were first obtained by splitting by barcode and removing the barcode and primers, followed by splicing of the R1 and R2 sequence data via FLASH software. The spliced tags were then quality controlled to obtain clean tags and chimera filtered to obtain effective tags for subsequent analysis, followed by noise reduction via the DADA2 method [24,25]. Each deduplicated sequence generated after noise reduction via DADA2 is referred to as an amplicon sequence variant (ASV) or feature sequence (corresponding to representative OTU sequences), and the table of the abundance of these sequences in the sample is referred to as the feature table (corresponding to the OTU table). The classify-sklearn algorithm of QIIME2 was used to annotate each ASV with a species via a pretrained naive Bayes classifier [26,27]. On the basis of the annotated results of the ASVs and the character tables of each sample, species abundance tables were generated at the kingdom, phylum, order, family, genus, and species levels, and these abundance tables with annotation information constituted the core of the amplicon analysis.

The microbial community compositions of the different samples were analyzed comparatively in terms of beta diversity. On the basis of the species annotation results and the abundance information of the characteristic sequences of all the samples, the characteristic sequences of the same classification were combined and processed to obtain the species profiling table. The phylogenetic relationships between the characteristic sequences were also used to further calculate the unweighted UniFrac (a kind of intersample distance calculation using the evolutionary information between microbial sequences in each sample; for more than one sample, a distance matrix was obtained) [28]. The unweighted UniFrac was further constructed using the abundance information of the signature sequences to construct weighted UniFrac distances [29]. Finally, the multidimensional data were downscaled via PCA to extract the most significant characteristic genera in each dataset and to best reflect the differences between the samples of each group [30]. The top 10 species, in terms of mean abundance at each taxonomic level, were selected from the three groups to generate a ternary plot to visualize the differences between the dominant species in the three groups at different taxonomic levels [31]. Differences among the three groups were analyzed via the Tukey and Kruskal–Wallis rank sum tests in the alpha diversity index analysis of intergroup variation, and box plots were generated [32].

## 3. Results

### 3.1. SQ Promotes Muscle Growth and Improves Immunity in Mice

SQ promoted muscle growth in mice. SQ at 15.6 mg/mL (the lowest effective concentration of SQ was determined by pretesting) was incubated with C2C12 mouse myoblasts for 12, 24, and 36 h, respectively. The results of CCK8 showed that SQ significantly increased the viability of C2C12 cells after 24 and 36 h of incubation (*p* < 0.05, Figure 1a). The EdU dye can be incorporated into replicating DNA, thus fluorescent dye exists in the nuclei of proliferating cells. As the results of the EdU assay show, compared with the control group, the red fluorescence represented by EdU in the SQ group cells increased significantly (*p* < 0.05), which indicates that SQ enhanced the proliferation ability of C2C12 cells (Figure 1b). These results suggest that SQ may promote the growth of mouse muscle cells *in vitro*. *In vivo*, SQ also increased the fiber diameter in mouse muscle, as shown by gavage administration to mice for 14 days (Figure 1c). We investigated the effects of SQ on myogenic regulators in muscle, and the results revealed that SQ significantly increased the mRNA expression of the myogenic regulators MYOG and MYF5 and inhibited the mRNA expression of MSTN (*p* < 0.05) but had no significant effect on MYoD (*p* > 0.05, Figure 1d). MYoD, MYOG, and MYF5 promote muscle proliferation and differentiation, whereas MSTN inhibits muscle proliferation and differentiation. These results suggest that SQ can promote muscle growth.

In addition, we investigated the effect of SQ on immunity. SQ was found to increase the phagocytic capacity of mouse RAW264.7 macrophages via the neutral red dye phagocytosis assay (Figure 1e). Both IL-1β and IL-2 are regulators of innate immunity, and their increased expression reflects increased immunity [33]. In mice, SQ also significantly promoted the mRNA expression of IL-1β and IL-2 in the mouse spleen (*p* < 0.05, Figure 1f) and improved the mouse splenic index (*p* < 0.05, Figure 1g) but had no significant effect on the mouse thymus (*p* > 0.05). In conclusion, these results suggest that SQ may improve the immunity of mice.

### 3.2. Stability of the CSEE

SQ has been shown to improve growth in mice, but there is a need to optimize SQ to minimize the feed-to-weight ratio. Therefore, a combination of SQ and CNS was used to supplement the diet. SQ was brown at room temperature, and CNS was bright yellow at room temperature (Appendix A). When the two solutions were mixed in different proportions, they formed a stable solution at room temperature, with the color becoming yellow as the proportion of hypericin increased and brown as the proportion of SQ increased. At 60 °C for 5 days, there was significant precipitation of SQ, and on day 10, the precipitation increased significantly, and the color of the solution gradually darkened (Appendix A), changing from brown to brownish black. The CSEE solution was relatively stable, with no obvious precipitation except for a slightly darker color. The results indicate that SQ is relatively stable at room temperature but is not resistant to high temperature, whereas the CNS and CSEE are more stable at room temperature and high temperature.

### 3.3. CSEE Improves the Growth Performance of Mice

The results after 14 days of treatment revealed that SQ had the greatest growth-promoting effect, with mice gaining the most weight within 14 days (Figure 2a–c). On day 14, the average weight gain of the mice in the SQ group was 16.58 g, whereas it was 13.3 g in the blank group. However, the daily food consumption of the mice in the SQ group was the highest (Figure 2d), suggesting that SQ may have led to weight gain by stimulating the mice to eat. The mice in the SQ group gained the most weight and had the lowest feed/gain ratio during the first 10 days, but on days 10–14, the CSEE-M and CSEE-L groups presented the greatest daily weight gain and the lowest feed/gain ratio (Figure 2c). These results suggest that, in the long term, CSEE-M and CSEE-L may improve feed conversion in mice more than SQ does. In addition, the CSEE group presented greater daily weight gain and a lower feed/gain ratio than the CNS group presented (Figure 2e,f). These results suggest that SQ is better than CNS at promoting growth and improving growth performance in mice but increases the rate of food consumption beyond 10 days. The optimized CSEE groups, CSEE-M and CSEE-L, had better weight gain and feed conversion than the CNS group and consumed less food than the SQ group.

### 3.4. CSEE Increases Mouse Vitality

During the 14-day feeding process, we unexpectedly discovered that some mice were more active, which was reflected in a greater willingness to move and a higher activity frequency. Therefore, we accurately evaluated the influence of different feeding conditions on the vitality of mice through TST and FST. The results of the TST revealed that the vitality of the mice in each group was significantly improved compared with that of the blank group (*p* < 0.05), while there was no significant difference between the experimental groups (Appendix A). The mean exercise times of the SQ, CNS, CSEE-H, CSEE-M, and CSEE-L groups were 275.29, 300, 296.44, 306.10, and 312.84 s, respectively. Both CSEE-M and CSEE-L were more effective than CNS in improving the vitality of the mice. The results of the FST revealed that, compared with the blank group, the CNS and CSEE groups presented significantly improved vigor (*p* < 0.05, Appendix A), whereas the SQ group presented no significant difference from the blank group (*p* > 0.05). All three CSEE treatments were more effective than CNS in increasing the vitality of the mice, with the CSEE-M group showing the best results. These results suggest that the CNS is more effective than the SQ in increasing mouse vitality, whereas the CSEE-M is superior to the CNS in increasing mouse vitality.

### 3.5. The Effects of CSEE on Blood Glucose and Lipid Serum Levels in Mice

To further investigate the mechanisms by which CSEEs improve growth performance in mice, we first considered glucose and lipid serum levels. Compared with those in the blank group, blood glucose concentrations were significantly greater in all experimental groups but were within the normal range of values (*p* < 0.05, Figure 3a), which may account for the high vitality of the mice in each experimental group (Appendix A). In terms of lipid metabolism, CSEE-H and CSEE-M significantly reduced TG levels compared with those in the blank group (*p* < 0.05, Figure 3b); CNS significantly reduced the levels of TC, LCL-C, and HDL-C (*p* < 0.05, Figure 3c–e), which may explain the slower weight gain in mice from the CNS and CSEE groups (Figure 1). Interestingly, SQ appeared to promote TBA secretion, whereas CNS had the opposite effect, reducing TBA levels (*p* > 0.05, Figure 3f). The TBA levels in all three CSEE groups were relatively lower than those in the blank group. In conclusion, the above results suggest that CSEEs can promote carbohydrate absorption and reduce lipid accumulation in mice.

### 3.6. Effect of CSEE on Digestive Enzymes

The activity of digestive enzymes is closely related to the growth performance of animals, and the better the digestion is, the greater the growth performance [34,35]. To this end, we further investigated the effects of CSEEs on digestive enzymes in the stomach and intestine of mice. Figure 4 shows the activities of α-amylase, lipase and protease in the stomach and intestine of mice 14 days after administration. The results showed that SQ significantly increased the activity of α-amylase in the stomach, whereas CNS inhibited the activity of α-amylase in the stomach (*p* < 0.05). In the duodenum, CSEE-M and CSEE-L significantly decreased the activity of lipase (*p* < 0.05), which may be the reason for the reduced lipid content in the blood of the mice in the CSEE-M and CSEE-L groups (Figure 3). Additionally, neither CSEE nor SQ has a significant influence on the activity of proteases in the intestinal tract.

### 3.7. CSEE Improves the Growth Performance and Slaughter Performance of Broilers

To investigate whether CSEEs can promote growth in broilers, the CSEE-M was selected for validation. The effects of CSEE-M on the growth performance and slaughter performance of broilers were investigated by adding 0.5–2% CSEE-M to broiler drinking water. In terms of survival rate, CSEE-M significantly increased the survival rate of broilers on days 14, 28, and 42 after administration (*p* < 0.05, Figure 5a,d,g), whereas there was no significant difference between the GAA group and the blank group (*p* > 0.05). In this study, the basal diet of the chickens did not contain any antibiotics. On the last day of the experiment, the survival rates of broilers in the CSEE-M group supplemented with 0.5%, 1%, and 2% drinking water were 72.22%, 91.67%, and 83.33%, respectively, whereas they were 53.60% and 61.11% in the blank control and GAA groups, respectively. In terms of mean body weight, there was no significant difference in body weight between the broiler groups at 14 days (*p* > 0.05, Figure 5b), and CSEE-M showed significant growth-promoting efficacy at 28 and 42 days (*p* < 0.05, Figure 5e,h). On day 28, the body weights of broilers in the CSEE-M group supplemented with 0.5%, 1%, and 2% drinking water were 37.21%, 23.40%, and 8.51% greater than those in the control group, respectively. On day 42, the body weights of broilers in the CSEE-M group supplemented with 0.5%, 1%, and 2% drinking water increased by 29.51%, 11.48%, and 8.20%, respectively, compared with those in the control group. In terms of the feed-to-gain ratio, there was no significant difference between the CSEE-M group and the blank group on day 42 (*p* > 0.05), but the GAA group presented a significantly greater feed-to-gain ratio than the blank group presented on day 14 (*p* < 0.05), suggesting that the addition of GAA to the diet on day 14 may increase feeding costs (Figure 5c,f,i).

We also investigated the effect of CSEE-M on broiler slaughter performance and observed no significant difference in the dressing percentage between the groups after 42 days of feeding (*p* > 0.05, Figure 5j) but a significant difference in muscle content (*p* < 0.05, Figure 5k). Compared with broilers on the control diet, feeding with 0.5% CSEE-M significantly increased the leg muscle content of broilers by approximately 16.12%, but feeding with 1% or 2% CSEE-M reduced the leg muscle content of broilers. These findings suggest that the effective concentration of CSEE-M should be 0.5% when increasing the muscle content of broilers.

### 3.8. CSEE-M Increases the Activity of Digestive Enzymes in the Broiler Intestine

CSEE-M significantly increased the activity of digestive enzymes in the intestines of broiler chickens. After 42 days of feeding, five broilers from each group were randomly slaughtered, and the duodenum was removed for digestive enzyme activity testing. Compared with the blank group, the low dose (0.5%) of CSEE-M increased the activity of amylase and lipase in the duodenum (Appendix A), but the difference was not significant (*p* > 0.05); the low dose (0.5%) of CSEE-M significantly increased the activity of protease in the broiler intestine (*p* < 0.01) (Appendix A), whereas the high and medium doses of CSEE-M had some inhibitory effects on intestinal digestive enzymes. The higher the concentration of CSEE-M was, the more pronounced the inhibitory effect was. In addition, the positive control drug GAA significantly reduced the activities of amylase, lipase, and protease in the broiler intestine compared with those in the control group (Appendix A). In conclusion, these results suggest that the effective concentration of CSEE-M in terms of promoting digestibility in broilers is 0.5%.

### 3.9. Effect of CSEE-M on the Species Richness of the Broiler Gut Microbiota

CSEE-M maintained the species richness of the broiler gut microbiota. As seen from the PCA plots, the blank control samples were relatively evenly distributed, whereas the samples from the 0.5% CSEE-M group started to cluster toward the middle, and the GAA group samples clustered more closely (Figure 6a). The ternary phase diagram at the phylum level reflected the dominant species between the groups, with Bacteroidota in the blank group, Verrucomicrobiota in the GAA group, and Chloroflexi in the 0.5% CSEE-M group (Figure 6b). The results for the four alpha diversity indices chao1, Simpson, Shannon, and pielou_e suggest that the gut microbiota richness of the broilers in the GAA group was significantly lower, whereas that of the broilers in the 0.5% CSEE-M group was not significantly different from that of the blank group (Figure 6c–f). In conclusion, these results suggest that GAA decreases the species richness of the broiler gut microbiota but that CSEEs have no significant effect on the species richness of the broiler gut microbiota.

We then further analyzed the differences in the gut microbiota composition between the sample groups at the genus level. At the genus level, CSEE-M significantly increased the richness of *Streptococcus* spp. in the broiler gut, but there was no significant difference at the species level compared with the blank group (Figure 6g,h). GAA significantly increased the richness of *Romboutsia* spp. and *Streptococcus* spp. in the broiler gut at the genus level but relatively decreased the richness of Lactobacillus spp. compared with both the blank and CSEE-M groups (Figure 6g,h). In addition, at the genus level, compared with GAA, CSEE-M relatively increased the abundance of Helicobacter and Achromobacter and relatively suppressed the abundance of Romboutsia (Figure 6g).

The gut microbiota also plays a key role in regulating the immunity and physiology of host animals [2]. To further elucidate the relationship between the gut microbiota and phenotype, a Venn diagram of the gut microbiota (genus level) and phenotype was drawn via Venny 2.1, and the microbiota unique to each group, as well as the phenotype, were analyzed (Appendix A). The results suggested that the downregulation of digestive enzyme activity and the increase in body weight in broilers in the GAA group could be related to the upregulation of *Romboutsia*. There are few studies on *Romboutsia* spp., and a recent report suggested that *Romboutsia* spp. may be associated with lipid metabolism [36]. Lactobacillus aviarius and Streptococcus are believed to improve the digestive ability and weight gain of broilers.

## 4. Discussion

The restricted utilization of AGPs in recent years has severely constrained the large-scale production of livestock and poultry. Consequently, there is an urgent necessity to develop new, safe, and inexpensive growth promoters. Furthermore, natural plant extracts have shown encouraging outcomes in promoting the growth of food animals, thereby attracting considerable research interest [37,38,39]. This study not only provides new alternatives for the application of growth promoters but presents new perspectives for research and development of growth promoters.

SQ is likely the most effective short-term growth promoter. In the short term, SQ can improve food utilization by increasing the activity of digestive enzymes and promoting animal growth. However, over time, SQ induces the conversion of excess ingested energy into fat storage, which might result in a fattening of the animal. In other words, SQ may be more appropriate for use in growing animals that have fast metabolism, are less inclined to fat storage, and may improve their growth rate, such as white-feathered broiler chickens, meat quails, Duroc pigs, etc. By contrast, for animals that require a higher percentage of fat, SQ is suitable for long-term use during the fattening period [40,41,42,43]. It has been reported that astragalus in SQ improves energy metabolism [44], which may be the reason why SQ leads to increased fat storage in animals. The capacity of SQ to increase digestive enzyme activity may be attributed to fried hawthorn. Previous studies have shown that fried hawthorn not only stimulates intestinal peristalsis and reduces the level of vasoactive intestinal peptide but increases digestive enzyme activity [12,45]. Finally, SQ is a promising growth promoter because of its simple preparation, straightforward administration, and low cost.

CSEE is also regarded as a highly promising long-term growth promoter. CNS is readily absorbed but may have a first-pass effect [46], and prolonged usage may reduce its growth-promoting effects. CNS lowers cholesterol levels and increases blood glucose levels, which may account for the increased vigor induced by CNS in mice. CSEE combines the advantages of both SQ and CNS to promote both growth and vigor in animals. Compared with SQ, CSEE not only enhanced animal feed conversion but was more stable than SQ. Among these, CSEE-M was the most effective in both increasing blood glucose levels and reducing lipid levels. We further speculate that CSEE-M may lead to weight gain by increasing muscle weight rather than fat accumulation. At present, most of the growth promoters we have explored promote animal growth by increasing energy metabolism, with little or no effect in promoting muscle growth [13]. Therefore, we believe that CSEE-M may be more suitable for food animals that require more muscle, such as broilers and beef cattle, which is why we ultimately chose broilers to evaluate the growth-promoting effects of CSEEs. In addition, as a liquid feed additive, CSEE-M can be added directly to the drinking water of livestock and poultry at low application rates, making it relatively inexpensive.

The formulation of natural plant extracts for growth promotion is likely to be more effective. In addition to antibiotics, natural plant extracts are the most abundant source of growth promoters. The addition of GAA to the diet significantly enhances muscle production and induces the conversion of the longest dorsal muscle to more oxidized myofibers in fattening pigs, thereby improving the growth performance of fattening pigs [37]. Inulin is a reserve polysaccharide widely distributed in plants of the Asteraceae, Platycodonaceae, and Gentianaceae families. When employed as a feed additive, inulin boosts growth performance and ameliorates carcass characteristics in fattening pigs [47]. The addition of sugar beet pulp to the diet improved growth performance and increased the lean meat percentage in meat ducks [48]. However, it is challenging to attain satisfactory growth promotion effects by using a single natural plant extract. Hence, we embraced the strategy of developing plant extract formulations In this study, the formulation composed of chao shanzha and huangqi extracts facilitated the proliferation of mouse muscle cells and increased the muscle content of the mice, which might be related to the CGA in hawthorn [37]. Additionally, SQ improved the immune function in mice, which may be the result of Astragalus exerting some efficacy [49]. In brief, SQ is more effective than a single extract because of its combination.

The enhancement in digestive enzyme activity can promote animal growth [50], thereby improving food utilization and making nutrient absorption more efficient. In mice, SQ showed efficacy in increasing gastric α-amylase activity, which is in line with previous studies indicating hawthorn amylase secretion and enzyme activity [45]. By contrast, CNS was revealed to be effective in inhibiting lipase activity and enhancing vitality in mice. This may be related to the presence of vitamin B1 and α-tocopherol in CNS. Moreover, the supplementation of additional vitamin B1 activates glycogen synthase kinase-3β (GSK-3β), thereby exerting an antidepressant effect [51]. Recent studies have indicated that α-tocopheryl acetate also has an antidepressant effect [52]. CSEE-M combines the advantages of SQ and CNS to increase the muscle ratio and improve digestive enzyme activity in both mice and broilers.

The relationships between growth promoters and the gut microbiota captured our attention. Although the association of GAA with Romboutsia spp. has not been reported, the ability of GAA to act as a precursor for creatine synthesis to increase growth performance and promote creatine metabolism has long been reported [53]. Whether this process could be mediated via Romboutsia also requires further investigated. Additionally, this study reports for the first time that GAA reduces the activity of intestinal digestive enzymes, but the mechanism remains unclear. Helicobacter is known as a group of pathogenic bacteria that cause diseases such as bacteraemia and gastric cancer [54,55]. However, it has also been reported that Helicobacter is negatively associated with reflux esophagitis, meaning that the incidence of reflux esophagitis increases after the eradication of Helicobacter [56,57,58]. Achromobacter is also a group of conditionally pathogenic bacteria that cause infections in immunocompromised patients [59,60]. The upregulation of Helicobacter and *Achromobacter* spp. in the gut may have increased the disease resistance of the chicks to some extent and would therefore have improved their survival rate. We hypothesize that the increase in muscle content and protease activity in broilers may be related to the supplementation of vitamins and amino acids by CNS, and the role of Helicobacter spp. and *Achromobacter* spp. in this process is unknown. The probiotics that maintain relatively high digestibility and promote growth in broilers are considered to be Lactobacillus aviarius and Streptococcus. Bacteria in the genus Lactobacillus have been reported to have anti-inflammatory [61,62] and insulin-promoting effects [63]. However, few studies have investigated Lactobacillus orientalis, which is believed to be an avian probiotic. *Streptococcus* spp. are both probiotic and pathogenic. For instance, *Streptococcus pyogenes* causes wound infection [64], rabbits die after *Streptococcus lactis* infection [65], and nutritional endocarditis in cattle is caused by *Streptococcus suis* infection [66]. Some beneficial streptococci, such as *Streptococcus thermophilus*, protect against acute cadmium toxicity in mice [67]. Nevertheless, the specific types of bacteria within the genus Streptococcus that are beneficial may require further validation.

In fact, we are highly interested in the components of SQ that promote muscle growth and enhance immunity, which will be further investigated in the future. Furthermore, the effects of CSEE-M and SQ on carcass performance and meat quality need to be further assessed. In addition, we propose that increasing digestive enzyme activity and improving the gut microbiota may promote animal growth. These are two novel viewpoints that deserve further investigation. Although there are few related reports and a lack of in-depth studies.

## 5. Conclusions

In conclusion, two growth promoters, SQ and CSEE-M, both based on chao shanzha and huangqi extracts, were developed in this study. SQ promoted muscle growth by improving glucose and lipid serum levels and enhancing immunity, thereby improving the growth performance of the animals. CSEE-M significantly improved the growth performance of both mice and broilers, increased the feed conversion ratio, markedly increased the activity of intestinal digestive enzymes in both mice and broilers, and maintained the diversity of the intestinal microbiota in broilers. Consequently, SQ and CSEE-M have great potential as growth promoters. Additionally, the promotion of muscle growth, the enhancement of digestive enzyme activity, and the improvement of the gut microbiota may be the focus for future development of novel growth promoters.

## Figures and Tables

**Figure 1 vetsci-11-00672-f001:**
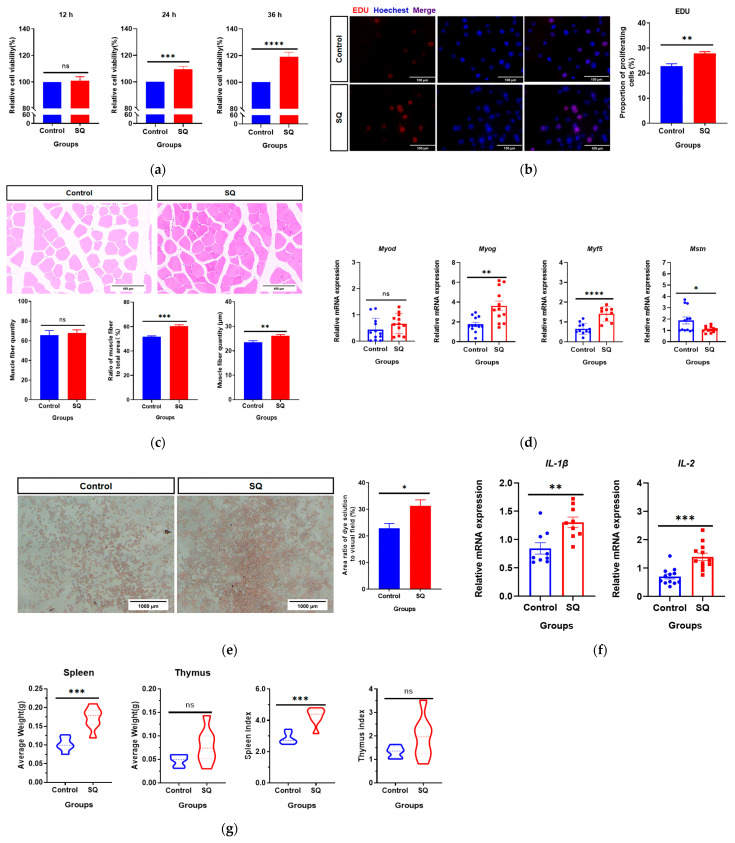
SQ promotes muscle growth and improves immunity in mice. (**a**) Effect of SQ on the viability of C2C12 cells. n = 6; (**b**) EdU assay. EdU binds to dividing nucleus and shows red fluorescence. Hoechst binds to all nuclei and shows blue fluorescence. Six fields of view were randomly selected to capture images for each experiment. Magnification: 400×. Scale bar: 100 μm; (**c**) Tissue sections of mouse leg muscles. Left leg muscles of mice were harvested for H&E staining. There were six mice from each group and one section was prepared for each mouse. Magnification: 40×. Scale bar: 400 μm; (**d**) mRNA expression of myogenic regulators in mouse muscle. n = 6; (**e**) Macrophage phagocytosis experiment. Macrophage phagocytosis is an epiphenomenon of innate immunity; the stronger macrophage phagocytosis can phagocytose, the more neutral red dye, the redder the cells. Magnification: 100×. Scale bar: 1000 μm; (**f**) mRNA expression of mesangial innate immunity factors in mouse spleen. n = 6; (**g**) Mouse spleen and thymus weights and spleen index and thymus index. n = 6. All cellular experiments as well as qPCR experiments were independently repeated three times. For all image data, six fields of view were randomly selected to capture images and show representative data. Analysis by *t* tests, ****: *p* < 0.0001; ***: *p* < 0.001; **: *p* < 0.01; *: *p* < 0.05; ns: not significant.

**Figure 2 vetsci-11-00672-f002:**
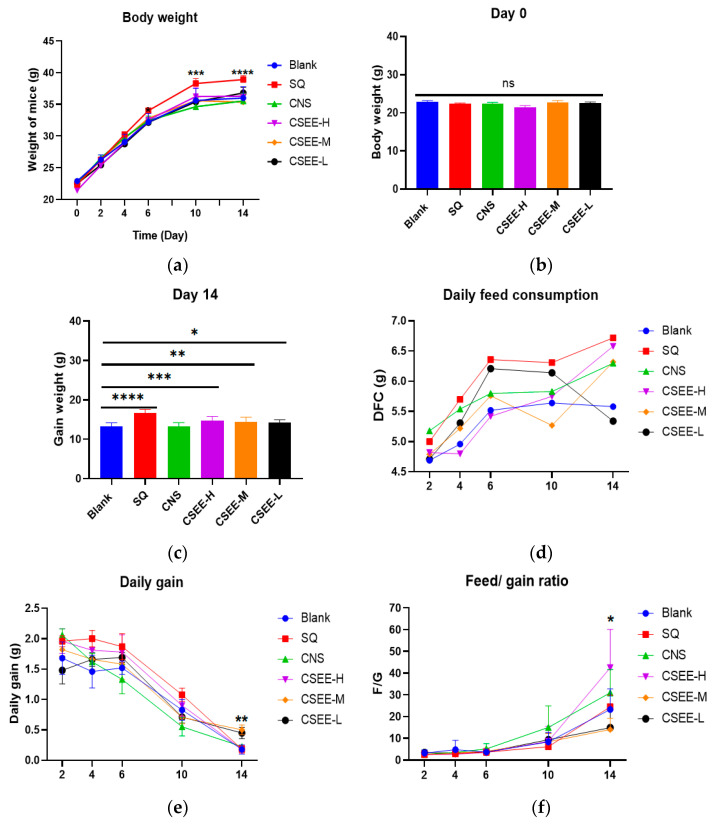
Growth performance of mice. (**a**) Changes in body weight of mice over fourteen days; (**b**) Body weight of mice in each group on day 0; (**c**) Body weight gain of mice in each group on day 14; (**d**) Changes in food consumption of mice over fourteen days; (**e**) Changes in daily gain of mice over fourteen days; (**f**) Changes in feed to gain ratio of mice over fourteen days. Blank, blank control; SQ, SQ group (orally administered at a dose of 3 g/kg); CNS, CNS group; CSEE-H, SQ:CNS = 1:5 (*v*/*v*); CSEE-M, SQ:CNS = 1:8 (*v*/*v*); CSEE, SQ:CNS = 1:10 (*v*/*v*). n = 10. Analysis via one-way ANOVA, ****: *p* < 0.0001; ***: *p* < 0.001; **: *p* < 0.01; *: *p* < 0.05; ns: not significant.

**Figure 3 vetsci-11-00672-f003:**
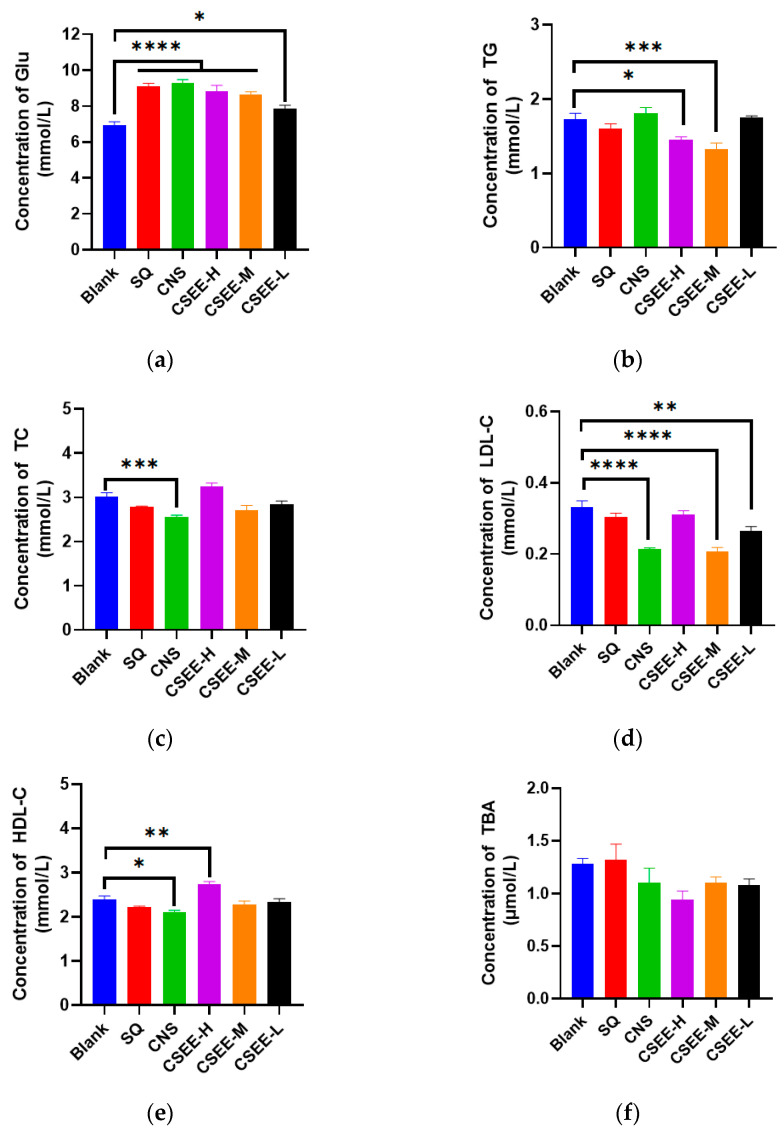
Serum biochemistry of mice 14 days after administration (**a**) Glu, blood glucose; (**b**) TG, triglycerides; (**c**) TC, total cholesterol; (**d**) LDL-C, Low-density lipoprotein cholesterol; (**e**) HDL-C, High-density lipoprotein cholesterol; (**f**) TBA, bile acids. blank, blank control; SQ, SQ group (dose administered at 3 g/kg); CNS, CNS group; CSEE-H, SQ:CNS = 1:5 (*v*/*v*); CSEE-M, SQ:CNS = 1:8 (*v*/*v*); CSEE-L, SQ:CNS = 1:10 (*v*/*v*). n = 10. Analysis via one-way ANOVA, ****: *p* < 0.0001; ***: *p* < 0.001; **: *p* < 0.01; *: *p* < 0.05. No column markers indicate no significant difference between the two groups.

**Figure 4 vetsci-11-00672-f004:**
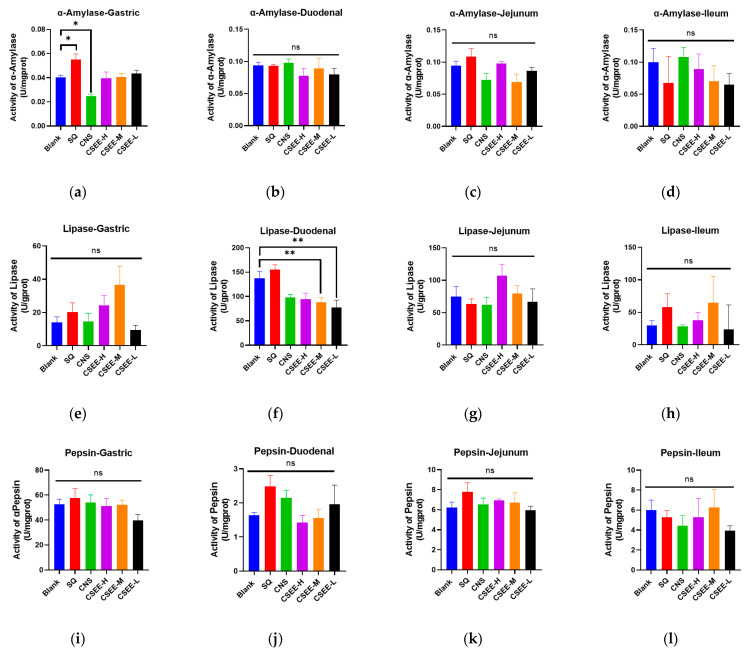
Changes in intestinal digestive enzymes in mice 14 days after administration. (**a**–**d**) The activity of α-amylase in the stomach, duodenum, jejunum and ileum; (**e**–**h**) Lipase activity in the stomach, duodenum, jejunum and ileum; (**i**–**l**) Protease activity in stomach, duodenum, jejunum and ileum. blank, blank control; SQ, SQ group (administered at 3 g/kg); CNS, CNS group; CSEE-H, SQ:CNS = 1:5 (*v*/*v*); CSEE-M, SQ:CNS = 1:8 (*v*/*v*); CSEE-L, SQ:CNS = 1:10 (*v*/*v*). n = 10. Analysis via one-way ANOVA, **: *p* < 0.01; *: *p* < 0.05; ns: not significant.

**Figure 5 vetsci-11-00672-f005:**
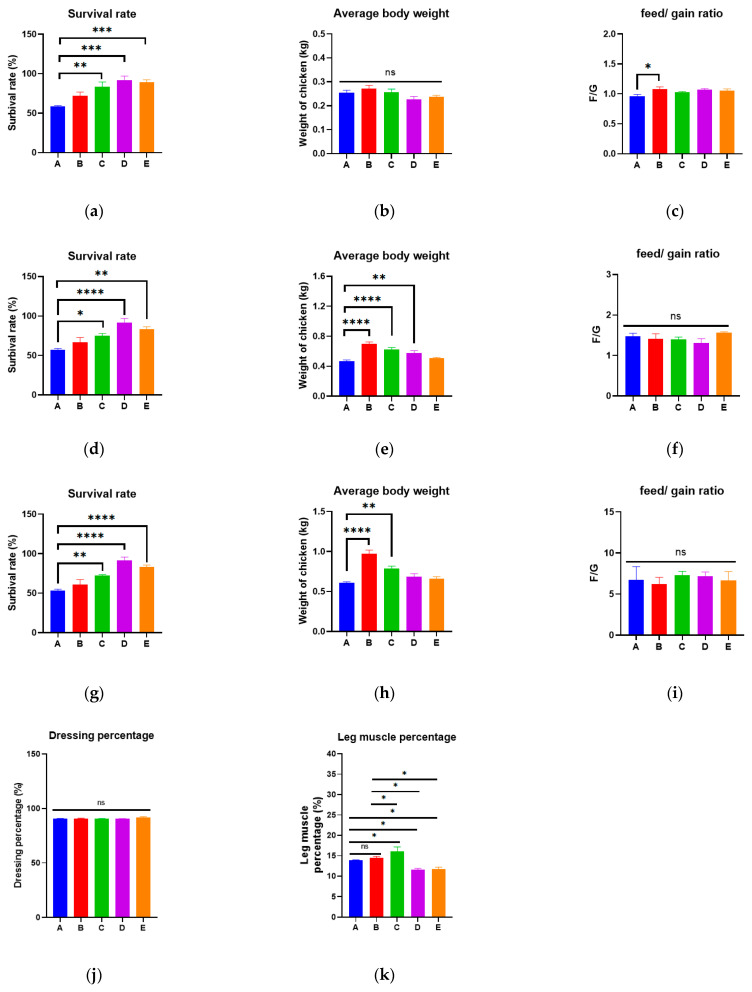
Growth and slaughter performance of broilers (**a**–**c**) Survival rate, average body weight and feed to gain ratio of broilers at 14 days; (**d**–**f**) Survival rate, average body weight and feed to gain ratio of broilers at 28 days; (**g**–**i**) Survival rate, average body weight and feed to gain ratio of broilers at 28 days; (**j**) Dressing percentage of broilers at 42 days of age; (**k**) Leg muscle content of broilers after 42 days of rearing. A, blank control; B, positive control (broiler diet supplemented with guanidinoacetic acid at 600 mg/kg); C, 0.5% CSEE-M; D, 1% CSEE-M; E, 2% CSEE-M. n = 10. Analysis via one-way ANOVA, ****: *p* < 0.0001; ***: *p* < 0.001; **: *p* < 0.01; *: *p* < 0.05; ns: not significant.

**Figure 6 vetsci-11-00672-f006:**
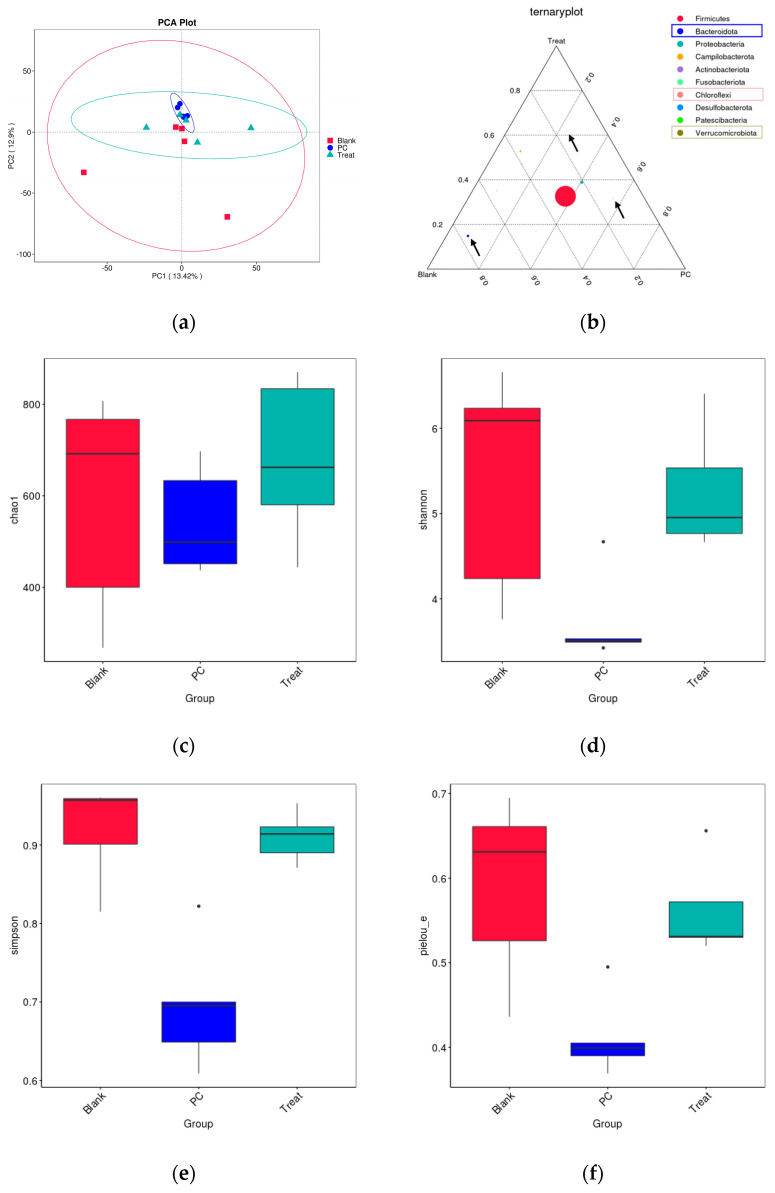
Species richness of broiler gut microbiota in different treatment groups. (**a**) Principal component analysis (PCA) plots; (**b**) Ternary phase diagram of the broiler gut microbiota at the phylum level for the three different treatment groups. The three vertices in the plot represent the three sample groups, the circles represent the species, and the size of circle is proportional to the relative abundance, the closer the circle is to a vertex, the more abundant the species is in that group of sample group; (**c**) The chao1 algorithm estimates the index of the number of OTU-containing elements in a community; (**d**) Shannon index; (**e**) Simpson’s index; (**f**) Uniformity index. n = 5. Abbreviations: Blank, blank control; PC, positive control (broiler diets supplemented with guanidinoacetic acid at 600 mg/kg); Treat, 0.5% CSEE-M group; (**g**) Differences in broiler gut microbiota at the genus level in each group; (**h**) Differences in broiler gut microbiota at the species level of in each group.

## Data Availability

All data generated or analyzed during this study are included in this published article [and its Appendix A]. The metagenomic sequencing data reported in this paper have been deposited in the Genome Sequence Archive [68], in the National Genomics Data Center [69], China National Center for Bioinformation/Beijing Institute of Genomics, Chinese Academy of Sciences (GSA: CRA012456) that are publicly accessible at https://ngdc.cncb.ac.cn/gsa/s/5H1674Ry (accessed on 1 September 2023).

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
