# Peer review of "Development and Evaluation of Non-Antibiotic Growth Promoters for Food Animals"

_vetsci, 2024, doi:10.3390/vetsci11120672_

Round 1

Reviewer 1 Report

Comments and Suggestions for Authors

Major comments

This study investigated the role of non-antibiotic growth promoter (NAGP) in promoting food animal growth. The authors found that a combination of hawthorn (also known as shanzha) and astragalus (also known as huangqi) extracts (SQ), and CSEE-M, have an important role in promoting animal growth. The SQ and CSEE-M involved in this study seem to have practical application value. The research content is very comprehensive. However, I still have some little issues that the authors need to address: 

Detail comments

1. In the abstract section, the author should supplement the research significance of SQ and CSEE-M in the last sentence.

2. The authors should add statistical analysis charts to Figures 1B and C.

3. The authors should not only draw conclusions about SQ promoting mouse immune function based on macrophage phagocytic function, transcription levels of IL-1β and IL-2, and changes in splenic index (Figure 1E-G).

4. Results 3.6. Please add data description about protease.

5. Results 3.7. Please explain why the initial mortality rate of the control group broiler chickens was so high (close to 50%).

6. The authors found that 0.5% CSEE-M can help increase muscle growth in broiler chickens and increase the activity of digestive enzymes. However, 1% CSEE-M or 2% CSEE-M can better reduce the mortality rate of broiler chickens. Should 1% or 2% concentration be given priority in practical applications?

7. I recommend the author to discuss the shortcomings of this study.

8. Pay attention to checking the entire text for any writing or formatting errors. For example, the comma in line 580 is repeatedly used.

Author Response

This study investigated the role of non-antibiotic growth promoter (NAGP) in promoting food animal growth. The authors found that a combination of hawthorn (also known as shanzha) and astragalus (also known as huangqi) extracts (SQ), and CSEE-M, have an important role in promoting animal growth. The SQ and CSEE-M involved in this study seem to have practical application value. The research content is very comprehensive. However, I still have some little issues that the authors need to address

Answer:

Thank you very much for your recognition of our research. This is the greatest encouragement for our work. Your suggestions are helpful for us to further improve this work. We express our sincere gratitude. The following are the point-to-point responses to the suggestions you put forward.

Point 1:

In the abstract section, the author should supplement the research significance of SQ and CSEE-M in the last sentence.

Answer:

Thank you very much for your suggestion. We have supplemented the abstract section, emphasizing the significance of developing SQ and CESS-M.

Point 2:

The authors should add statistical analysis charts to Figures 1B and C.

Answer:

Your suggestion is highly commendable. We hold the view that the mutual substantiation of quantitative analysis and image results can better elucidate the issue. Hence, not only quantitative analyses were added to FIG 1B and FIG 1C, but also a quantitative analysis of FIG 1E was supplemented. Additionally, we discovered that some of the images in FIG 1 of the revised manuscript were not fully displayed, and we have accordingly made adjustments.

Point 3:

The authors should not only draw conclusions about SQ promoting mouse immune function based on macrophage phagocytic function, transcription levels of IL-1β and IL-2, and changes in splenic index (Figure 1E-G).

Answer:

Your suggestion is highly constructive. Although our results have provided evidence that SQ enhances immunity, the in vitro results alone are insufficient to clarify this issue, and our summary of the results might have been too absolute. Here, we have modified the final conclusion to "In conclusion, these results suggest that SQ may improve the immunity of mice" (line 357-358).

Point 4:

Results 3.6. Please add data description about protease.

Answer:

We are grateful for your suggestion. We have supplemented the description of the protease results in Section 3.6 of the results at lines 459 - 460.

Point 5:

Results 3.7. Please explain why the initial mortality rate of the control group broiler chickens was so high (close to 50%).

Answer:

The questions you raised are highly significant. In this study, the feed we provided to the chicks was free of any antibiotics. Its main components include: Corn, soybean meal, puffed soybeans, corn protein meal, stone meal, DL-methionine, L-lysine sulphate, vitamins, sodium chloride, etc. (More detailed information is presented in Table S3). We also emphasized in line 474 of the results section: "In this study, the basal diet of the chickens did not contain any antibiotics". Generally, there is a certain natural mortality rate for broiler chicks during the brooding period without the protection of antibiotics. Our research results indicate that after adding CSEE-M, the chicks can maintain a survival rate close to 100% even without antibiotics, which precisely provides a strong basis for the potential of CSEE-M as an antibiotic substitute.

Point 6:

The authors found that 0.5% CSEE-M can help increase muscle growth in broiler chickens and increase the activity of digestive enzymes. However, 1% CSEE-M or 2% CSEE-M can better reduce the mortality rate of broiler chickens. Should 1% or 2% concentration be given priority in practical applications?

Answer:

The question you raised is extremely excellent. Continuing from the reply we gave in Point 5. In actual production, most feeds contain antibiotics. Therefore, on the premise of ensuring the survival rate of chicks, the concentration we recommend is 0.5%. However, if considering the complete substitution of antibiotics, a concentration of 1% would be more appropriate, although this might sacrifice a little of the gain from the slaughter rate.

Point 7:

I recommend the author to discuss the shortcomings of this study.

Answer:

Thank you very much for your suggestion. In fact, regarding the shortcomings of this study, we have elaborated in the discussion section. We first discussed the advantages of SQ and CESS-M, and of course, also mentioned the deficiencies in the research on them. Additionally, regarding the areas that still lack further research in this study, we have summarized them in the last paragraph of the discussion section (lines 659 - 665).

Point 8:

Pay attention to checking the entire text for any writing or formatting errors. For example, the comma in line 580 is repeatedly used.

Answer:

We are extremely grateful for your sincere suggestions. We have also identified some formatting errors throughout the text and have carried out a meticulous revision anew.

Reviewer 2 Report

Comments and Suggestions for Authors

1. CNS is generally known in the world as central nervous system, so it should be changed.

2. There are editorial errors on figures e.g. in line 327 is 400mcm, and on picture is 1000mcm. Find other errors.

3. Description of columns on fig. 5 should be similar to previous ones.

Author Response

Point 1:

CNS is generally known in the world as central nervous system, so it should be changed.

Answer:

Thank you very much for your suggestion. I have considered your concern regarding the abbreviation 'CNS' being commonly known as 'Central Nervous System'. However, in my research and throughout the paper, 'CNS' is consistently used in a specific and well-defined context that is distinct from the common meaning. I have ensured that the meaning of 'CNS' is clearly explained and defined within the paper to avoid any confusion for the readers. I sincerely hope this explanation is satisfactory and that the continued use of 'CNS' in this specific sense does not cause any issues for the understanding and interpretation of the research presented.

Point 2:

There are editorial errors on figures e.g. in line 327 is 400mcm, and on picture is 1000mcm. Find other errors.

Answer:

Thank you very much for your suggestion. We carefully examined the description of FIG 1B in lines 327 to 328 and found that there should be no problem. Our field magnification was 400× and the scale bar was 100 μm. Additionally, upon careful inspection of this section, we discovered that the problem lay in the description of FIG 1E: "1000×. Scale bar: 400 μm". We have meticulously revised this error and changed it to "100×. Scale bar: 1000 μm" (lines 367 - 368). Furthermore, we also checked the other figure captions throughout the text and revised the erroneous parts. In conclusion, we are extremely grateful for your suggestion.

Point 3:

Description of columns on fig. 5 should be similar to previous ones.

Answer:

Thank you very much for your suggestion! We noticed that the image layout of FIG 5 was not consistent with the description in the results. In the results, it was described that the first column was the survival rate and the second column was the average body weight, but in the picture, the first column presented was the average body weight and the second column was the survival rate. This might be an error caused by the layout. We are extremely grateful to you for pointing out this mistake, which will greatly assist us in perfecting this paper. Regarding this error, we have adjusted the layout in FIG 5 and swapped the pictures in the first and second columns.

Reviewer 3 Report

Comments and Suggestions for Authors

The paper is well-written, and the topic is highly relevant. I suggest enhancing the description in the Materials and Methods section regarding the techniques used for muscle analysis, as well as expanding on the related results

Author Response

The paper is well-written, and the topic is highly relevant. I suggest enhancing the description in the Materials and Methods section regarding the techniques used for muscle analysis, as well as expanding on the related results.

Answer:

Thank you very much for your recognition of our work, which has greatly encouraged us! Regarding your suggested description of the muscle analysis method, we have supplemented it in Section 2.6 of the Materials section (lines 164 - 166). In the Results section, we have also supplemented the quantitative analysis of the muscles in FIG 1C. Thank you again for your suggestion!

Reviewer 4 Report

Comments and Suggestions for Authors

General comments:

I think the work presented in the paper is relevant to the field and brings novelty. However, the manuscript is poorly written and must be revised by a language expert. A significant number of confusing sentences and misspellings make it difficult to understand what the authors are trying to communicate. Figures, also, need to be revised for misspellings.

Additionally, there is a concern about several of the assays utilized in this study, such as the “forced swimming experiment” and “tail suspension test” which cause unnecessary stress to the animals, when more simple techniques as open field locomotor test and rotarod test could have been used and achieved the same results or even better results.

The material and methods section does not allow the replication of the study, because the assays are poorly described and language is broken and full of grammar mistakes. Another language issue throughout the manuscript the language used to describe cells is used to describe animals and most of the time it is not applicable (e.g.: viability of the mice, motility of the mice).

Specific comments:

Lines 12-14: Sentence is disconnected from the previous. It is suggested to change to: "However, it has been associated with increasing antibiotic resistance.."

Line 19: The “CSEE-M” acronym appears in the abstract without any explanation of what the acronym means. Add a proper description in the abstract.

Lines 95 – 101: Substitute to: "The glucose (Glu), triglyceride (TG), total cholesterol (TC), low-density lipoprotein cholesterol (LDL-C), high-density lipoprotein cholesterol (HDL-C), and total bile acid (TBA) test kits were purchased from Shenzhen Myri Biomedical Electronic Co., Ltd. The pepsin (A080-1-1), alpha-amylase (C016-1-1), and lipase (A054-1-1) test kits were purchased from Nanjing Jiancheng Technology Co., Ltd."

Sections 2.3, 2.6, 2.7: Please, provide a brief description of the methods. English revision is required.

Section 2.4: Is this a proliferation assay or a migration assay? EdU assay is to measure the cell proliferation, there is no need for a cell crawler in the plate. What was the need for the cell crawler?

Lines 115-116: Add more details to the methods: How long the cells were incubated with the treatments? What was the concentration of extracts used for each treatment?

Lines 127-128: Please, review this sentence.

Section 2.5: Broken English makes it difficult to understand the methodology, it needs to be re-written.

Lines 184-186: Why only the CSEE groups were evaluated for the growth-promoting performance? Please, revise the text.

Section 2.12 and 2.13: This is a nutritional study, what was the benefit of applying these techniques? The treatments are supposed to act as probiotics and not drugs.

Line 197: “crawl on their tails” Do the authors mean tail climbing?

Line 222: 10 broiler chickens from each replica? Please, add further details of the sample collection.

Lines 228-229: Why only 3 out of 5 groups were selected for this analysis?

Lines 290-293: This paragraph should be the first of this section. Additionally, it should be re-written as follows: "All the statistical analyses in this study, except for the macrogenomic dataset, were performed via GraphPad Prism 8. Analysis by t-tests (Student’s t-tests) or one-way ANOVA (add what results were analyzed by t-test or ANOVA). All the results were considered significantly different when the P value was less than 0.05.ns: P >0.05; *: P<0.05; **: P<0.01; ***: P<0.001;****: P<0.0001."

Line 296: “SQ promoted growth in mice” should be removed.

Line 299: What assay was done to test viability?

Line 300: “may promote the growth of mouse muscle cells” Viability is the capacity of the cell to stay a live, it does not corelates with proferation or growth. Please, change this statement to match with the data.

Line 301: “EDU” should be substitute as “EdU”.

Lines 302-303: It is suggested to add a transition between describing the in vitro and the vivo data, with a brief explanation of what procedure was done to the mice. Additional information needs to be added: What assay was done to evaluate the muscle fiber diameter? How it was quantified?

Lines 308-309: The body weight data wasn't shown at this point. This conclusion can't be drawn based on the data presented.

Lines 312-313: “Both IL-1β and IL-2 are regulators of innate immunity, and their increased expression reflects increased immunity.”  Authors should add literature references to this statement.

Line 315: Add to material and methods how the spleen and the thymus indexes was calculated.

Line 316: Thymus tissue collection was not mentioned in the methodology, please, add to it.

Lines 316-317: There is no immune cell quantification, only the increase of 2 Interleukines, this data isn't enough to conclude an increase in immune system function. Please, modify the conclusion or add more data to support this conclusion.

Line 342: “slight increase in color” Not sure what the authors mean by increase in color. Please, further explain.

Lines 358-360: Rewrite to: "The optimized CSEE groups CSEE-M and CSEE-L had better weight gain and feed conversion than the CNS group and consumed less food than SQ group." Growth implies the mice were bigger in length, but the authors did not present this data only body weight.

Lines 371-372: Common locomotor tests that you cause less stress could have been used as the open field locomotor activity test and the rotarod test. Why authors decide to use 2 tests that cause unnecessary stress?

Lines 381 and 383: What do authors mean by viability of the mice? Do authors mean vitality?

Section 3.5: Serum chemistry analysis, only measures the concentration of glucose and lipids in the blood, it isn't used to evaluate glycolipid metabolism. This section needs to be reviewed for English and scientific content. Glycolipids are lipids that contain carbohydrates attached by a glycosidic bond and are essential components of cell membranes.

Line 387: Authors should include a line, saying what analysis was performed, before writing the results.

Line 446: What the authors mean for slaughter rate. Please, revise the text.

Comments on the Quality of English Language

The manuscript is poorly written, and must be revised by a language expert. A significant number of confusing sentences and misspellings make it difficult to understand what the authors are trying to communicate.

Author Response

I think the work presented in the paper is relevant to the field and brings novelty. However, the manuscript is poorly written and must be revised by a language expert. A significant number of confusing sentences and misspellings make it difficult to understand what the authors are trying to communicate. Figures, also, need to be revised for misspellings.

The material and methods section does not allow the replication of the study, because the assays are poorly described and language is broken and full of grammar mistakes. Another language issue throughout the manuscript the language used to describe cells is used to describe animals and most of the time it is not applicable (e.g.: viability of the mice, motility of the mice).

Answer:

Thank you very much for your recognition of our work, which is a tremendous encouragement for us! Regarding the language issues you mentioned, it needs to be stated that after the completion of our manuscript, we sought the language polishing service from a professional translation company, Proof Reading Service.com Ltd. Nevertheless, it is undeniable that during the subsequent continuous revision process of the paper, it passed through the hands of different people, and due to the different language habits of each person, some language problems emerged. Your suggestions on the language aspect are all very good, and we have basically adopted all of them. For the questions in doubt, we will reply point by point in the following responses. Here, we thank you again for your suggestions, which have greatly helped us improve this work!

Point 1:

Additionally, there is a concern about several of the assays utilized in this study, such as the “forced swimming experiment” and “tail suspension test” which cause unnecessary stress to the animals, when more simple techniques as open field locomotor test and rotarod test could have been used and achieved the same results or even better results.

Answer:

Your suggestion is excellent! Firstly, both the "forced swimming experiment" and the "tail suspension test" are highly classic, and they are still in use even nowadays (see the references below), which indicates their credibility. Secondly, the advantage of these two experiments lies in the ability to assess the locomotor ability and mental status of animals through simple conditions, while the "open field locomotor test" and the "rotarod test" might require the preparation of additional detection or analysis equipment. Although the operation is simpler, the requirements for the laboratory are higher. Finally, it needs to be declared that our initial focus was on the muscles and locomotor ability of the animals. It was only by chance that we discovered that the mice taking CSEE were more energetic and displayed greater activity. For this phenomenon, we only intended to conduct a preliminary assessment as a supplementary result of CSEE's improvement of the locomotor ability of animals, and thus did not conduct a further systematic exploration.

  • Chen Z, Gu J, Lin S, et al. Saffron essential oil ameliorates CUMS-induced depression-like behavior in mice via the MAPK-CREB1-BDNF signaling pathway. J Ethnopharmacol. 2023;300:115719. doi:10.1016/j.jep.2022.115719.
  • Wang JY, Zhang Y, Chen Y, et al. Mechanisms underlying antidepressant effect of transcutaneous auricular vagus nerve stimulation on CUMS model rats based on hippocampal α7nAchR/NF-κB signal pathway. J Neuroinflammation. 2021;18(1):291. Published 2021 Dec 17. doi:10.1186/s12974-021-02341-6.
  • Ma J, Wang R, Chen Y, Wang Z, Dong Y. 5-HT attenuates chronic stress-induced cognitive impairment in mice through intestinal flora disruption. J Neuroinflammation. 2023;20(1):23. Published 2023 Feb 3. doi:10.1186/s12974-023-02693-1.
  • Zhou L, Liu D, Xie Z, et al. Electrophysiological Characteristics of Dorsal Raphe Nucleus in Tail Suspension Test. Front Behav Neurosci. 2022;16:893465. Published 2022 May 31. doi:10.3389/fnbeh.2022.893465.

Point 2:

Lines 12-14: Sentence is disconnected from the previous. It is suggested to change to: "However, it has been associated with increasing antibiotic resistance.."

Line 19: The “CSEE-M” acronym appears in the abstract without any explanation of what the acronym means. Add a proper description in the abstract.

Lines 95 – 101: Substitute to: "The glucose (Glu), triglyceride (TG), total cholesterol (TC), low-density lipoprotein cholesterol (LDL-C), high-density lipoprotein cholesterol (HDL-C), and total bile acid (TBA) test kits were purchased from Shenzhen Myri Biomedical Electronic Co., Ltd. The pepsin (A080-1-1), alpha-amylase (C016-1-1), and lipase (A054-1-1) test kits were purchased from Nanjing Jiancheng Technology Co., Ltd."

Answer:

Thank you very much for your suggestions. They are all excellent and revisions have been made in accordance with your advice.

Point 3:

Sections 2.3, 2.6, 2.7: Please, provide a brief description of the methods. English revision is required.

Answer:

Thank you very much for your suggestion. We have supplemented Sections 2.3, 2.6, and 2.7 in greater detail.

Point 4:

Section 2.4: Is this a proliferation assay or a migration assay? EdU assay is to measure the cell proliferation, there is no need for a cell crawler in the plate. What was the need for the cell crawler?

Answer:

Thank you very much for pointing out this issue. In fact, the reason we employed cell climbing slices for the EdU assay is that it enables the capture of images under an upright microscope to obtain higher-resolution pictures. Of course, this experiment can also be conducted directly in the well plate instead of using the climbing slices. Additionally, we found that in some published literature, "cell crawler" (see References 1 - 3) and "Cell climbing slice" (see References 4 - 6) might refer to the same object, namely the prototypical glass slide for cell colonization, which is used to prepare cell sections after cell staining. Here, we have decided to change "cell crawler" to "cell climbing slice".

  • Wang S, Wang Y, Hu X, et al. Dermal FOXO3 activity in response to Wnt/β-catenin signaling is required for feather follicle development of goose embryos (Anser cygnoides). Poult Sci. 2024;103(3):103424. doi:10.1016/j.psj.2024.103424
  • Xi Y, Pan Y, Li M, Zeng Q, Wang M. Evaluation of the application potential of Bdellovibrio sp. YBD-1 isolated from Yak faeces. Sci Rep. 2024;14(1):13010. Published 2024 Jun 6. doi:10.1038/s41598-024-63418-9
  • Si X, Wang X, Wu H, et al. Inhibition Effect of STING Agonist SR717 on PRRSV Replication. Viruses. 2024;16(9):1373. Published 2024 Aug 29. doi:10.3390/v16091373
  • Dong X, Shu L, Zhang J, et al. Ogt-mediated O-GlcNAcylation inhibits astrocytes activation through modulating NF-κB signaling pathway. J Neuroinflammation. 2023;20(1):146. Published 2023 Jun 22. doi:10.1186/s12974-023-02824-8
  • Dong S, Liang S, Cheng Z, et al. ROS/PI3K/Akt and Wnt/β-catenin signalings activate HIF-1α-induced metabolic reprogramming to impart 5-fluorouracil resistance in colorectal cancer. J Exp Clin Cancer Res. 2022;41(1):15. Published 2022 Jan 8. doi:10.1186/s13046-021-02229-6
  • Cheng XJ, Guan FL, Li Q, Dai G, Li HF, Li XK. AlCl3 exposure regulates neuronal development by modulating DNA modification. World J Stem Cells. 2020;12(11):1354-1365. doi:10.4252/wjsc.v12.i11.1354

Point 5:

Lines 115-116: Add more details to the methods: How long the cells were incubated with the treatments? What was the concentration of extracts used for each treatment?

Lines 127-128: Please, review this sentence.

Section 2.5: Broken English makes it difficult to understand the methodology, it needs to be re-written.

Answer:

Thank you very much for your suggestion. In Section 2.4, we have replenished the details of the concentration and time of SQ treatment of C2C12 cells anew (lines 133 - 134); for Sections 2.4 and 2.5, we have re-reviewed and revised to make the language more in line with the academic style.

Point 6:

Lines 184-186: Why only the CSEE groups were evaluated for the growth-promoting performance? Please, revise the text.

Section 2.12 and 2.13: This is a nutritional study, what was the benefit of applying these techniques? The treatments are supposed to act as probiotics and not drugs.

Line 197: “crawl on their tails” Do the authors mean tail climbing?

Line 222: 10 broiler chickens from each replica? Please, add further details of the sample collection.

Lines 228-229: Why only 3 out of 5 groups were selected for this analysis?

Answer:

Thank you very much for your suggestion. The description in Section 2.11 has been revised (lines 216 - 218); the reason for adopting the two experiments of TST and FST in 2.12 and 2.13 is that during the animal experiments, we found that the mice taking CSEE were more active and energetic. We believe that these two experiments can serve as supplements for evaluating the impact of CSEE on the locomotor ability of mice; the expression "crawl on their tails" is indeed inappropriate, and we have revised it in lines 230 - 232; we have provided a more detailed and accurate description of the grouping of broilers (lines 248 - 449); among the five groups of broilers, we analyzed the indicators such as intestinal digestive enzymes and slaughter performance of all groups. These results indicated that 0.5% CSEE-M had a better growth-promoting effect. Therefore, in the subsequent intestinal flora detection, we only selected the 0.5% CSEE-M treatment group, the blank group, and the positive control group for further analysis.

Point 7:

Lines 290-293: This paragraph should be the first of this section. Additionally, it should be re-written as follows: "All the statistical analyses in this study, except for the macrogenomic dataset, were performed via GraphPad Prism 8. Analysis by t-tests (Student’s t-tests) or one-way ANOVA (add what results were analyzed by t-test or ANOVA). All the results were considered significantly different when the P value was less than 0.05.ns: P >0.05; *: P<0.05; **: P<0.01; ***: P<0.001;****: P<0.0001."

Line 296: “SQ promoted growth in mice” should be removed.

Line 299: What assay was done to test viability?

Line 300: “may promote the growth of mouse muscle cells” Viability is the capacity of the cell to stay a live, it does not corelates with proferation or growth. Please, change this statement to match with the data.

Line 301: “EDU” should be substitute as “EdU”.

Lines 302-303: It is suggested to add a transition between describing the in vitro and the vivo data, with a brief explanation of what procedure was done to the mice. Additional information needs to be added: What assay was done to evaluate the muscle fiber diameter? How it was quantified?

Lines 308-309: The body weight data wasn't shown at this point. This conclusion can't be drawn based on the data presented.

Lines 312-313: “Both IL-1β and IL-2 are regulators of innate immunity, and their increased expression reflects increased immunity.”  Authors should add literature references to this statement.

Answer:

Thank you very much for your suggestion. We have adopted the revision suggestions for Section 2.16; the sentence "SQ promoted growth in mice" has been changed to "SQ promoted muscle growth in mice" (line 337). The writing of our results section is in the structure of "general - specific - general", which can greatly enhance the readability of the article and enable readers to quickly obtain the information of the results; lines 339 - 340, we have supplemented "CCK8" to make this sentence more accurate; the sentence "may promote the growth of mouse muscle cells" is indeed inappropriate to be placed behind the results of CCK8. We have adjusted it and placed it behind the results of EdU (lines 341 - 343); all the non-standard "EDU" have been modified to "EdU"; the transitional description between in vivo and in vitro experimental results has been supplemented (lines 341 - 345). Additionally, we have also supplemented the quantitative analysis of muscle tissue (FIG 1C); the description of "body weight" is not appropriate here, and we have modified this sentence (lines 350 - 351); the references regarding "IL-1β and IL-2" have also been supplemented (line 355).

Point 8:

Line 315: Add to material and methods how the spleen and the thymus indexes was calculated.

Line 316: Thymus tissue collection was not mentioned in the methodology, please, add to it.

Lines 316-317: There is no immune cell quantification, only the increase of 2 Interleukines, this data isn't enough to conclude an increase in immune system function. Please, modify the conclusion or add more data to support this conclusion.

Line 342: “slight increase in color” Not sure what the authors mean by increase in color. Please, further explain.

Lines 358-360: Rewrite to: "The optimized CSEE groups CSEE-M and CSEE-L had better weight gain and feed conversion than the CNS group and consumed less food than SQ group." Growth implies the mice were bigger in length, but the authors did not present this data only body weight.

Answer:

Thank you very much for your suggestions, which are all very good. According to your suggestions, in Section 2.11 of the Materials section, we supplemented the description of thymus and spleen sampling and the formula for calculating the thymus/spleen index (lines 218 - 220); in FIG 1E, we supplemented the quantitative results of macrophage phagocytosis of neutral red dye; the expression "slight increase in color" was inaccurate, and we have modified it to "The CSEE solution was relatively stable, with no obvious precipitation except for a slightly darker color" (lines 383 - 384); "The optimized CSEE groups CSEE-M and CSEE-L had better weight gain and feed conversion than the CNS group and consumed less food than SQ group" Thank you for your modification suggestion, and we have adopted it (lines 402 - 404).

Point 9:

Lines 371-372: Common locomotor tests that you cause less stress could have been used as the open field locomotor activity test and the rotarod test. Why authors decide to use 2 tests that cause unnecessary stress?

Lines 381 and 383: What do authors mean by viability of the mice? Do authors mean vitality?

Section 3.5: Serum chemistry analysis, only measures the concentration of glucose and lipids in the blood, it isn't used to evaluate glycolipid metabolism. This section needs to be reviewed for English and scientific content. Glycolipids are lipids that contain carbohydrates attached by a glycosidic bond and are essential components of cell membranes.

Line 387: Authors should include a line, saying what analysis was performed, before writing the results.

 Line 446: What the authors mean for slaughter rate. Please, revise the text.

Answer:

Thank you very much for your suggestions. Firstly, regarding the FST and TST experiments, we have provided explanations in the responses of Point 1 and Point 6. Both the FST and TST experiments are highly classic and are still in use even nowadays, which indicates their compliance with animal ethics. Moreover, the advantage of these two experiments lies in the ability to assess the locomotor ability and mental status of animals through simple conditions, while the "open field locomotor test" and the "rotarod test" might require the preparation of additional detection or analysis equipment. Although the operation is simpler, the requirements for the laboratory are higher. The description of "viability" was inaccurate, and we have replaced it with "vitality"; the description of "glycolipid metabolism" was ambiguous and has been changed to "glucose and lipid metabolism"; the first sentence of Section 3.4 has been revised (lines 413 - 416); the expression "slaughter rate" was inaccurate and has been changed to "Dressing percentage".

Round 2

Reviewer 4 Report

Comments and Suggestions for Authors

General comments:

This reviewer thinks the work presented in the paper is relevant to the field and brings novelty. The authors did a great job addressing all the comments and doing most of the requested modifications required by the reviewer.

Specific comments:

Lines 162-163: Substitute "... and photography." for "... and picture were taken."

Lines 169-170: Please, add further details, how many fields were evaluated? which magnification was used? Who did the evaluation (board-certified pathologist or trained scientist)?

Line 224: Substitute "/" for "or", it reads very confusingly using the current punctuation.

Line 225: Does "weight of mice" mean "body weight"? If so, please, modify it.

Line 360: EdU dye is incorporated in the replicating DNA, and because of that proliferating cells present fluorescent dye in their nucleus. Please, correct the concept in the text.

Section 3.5: Quantification of glucose and lipids in the blood serum does not measure their metabolism; it only shows an increase in the absorption/production or decreased excretion of these analytes. Please correct this in the manuscript.

Line 662: Please, substitute "glucose and lipid metabolism" to "glucose and lipid serum levels".

Author Response

The following is a point-to-point response to the reviewer’s comments.

Reviewer #1:

This reviewer thinks the work presented in the paper is relevant to the field and brings novelty. The authors did a great job addressing all the comments and doing most of the requested modifications required by the reviewer.

Answer:

Thank you very much for your recognition of our research. This is the greatest encouragement for our work. Your suggestions are helpful for us to further improve this work. We express our sincere gratitude. The following are the point-to-point responses to the suggestions you put forward.

Point 1:

Lines 162-163: Substitute "... and photography." for "... and picture were taken."

Lines 169-170: Please, add further details, how many fields were evaluated? which magnification was used? Who did the evaluation (board-certified pathologist or trained scientist)?

Line 224: Substitute "/" for "or", it reads very confusingly using the current punctuation.

Line 225: Does "weight of mice" mean "body weight"? If so, please, modify it.

Answer:

Thank you very much for your suggestion. We have revised the expression "... and photography." in lines 164 - 165 and changed it to "... and picture were taken."; In lines 170 - 173, we provided a more detailed description of the details of obtaining the pathological section images, and the analysis of this section was conducted by professional pathological laboratory technicians; In line 224, the only description with "/" is "feed  : gain ratio (F/G)", and we believe that F/G is a very classic description method in the field of feed research and does not require revision. Additionally, if it refers to the inappropriateness of "pleen/thymus" in lines 228 - 229, we have now changed it to "pleen or thymus"; and changed "weight of mice" in the formula "The spleen or thymus index of mice = ((weight of spleen or thymus) / weight of mice) * 100%." to "body weight of mice" (lines 228 - 229).

Point 2:

Line 360: EdU dye is incorporated in the replicating DNA, and because of that proliferating cells present fluorescent dye in their nucleus. Please, correct the concept in the text.

Section 3.5: Quantification of glucose and lipids in the blood serum does not measure their metabolism; it only shows an increase in the absorption/production or decreased excretion of these analytes. Please correct this in the manuscript.

Line 662: Please, substitute "glucose and lipid metabolism" to "glucose and lipid serum levels".

Answer:

Your suggestion is highly commendable. We have carefully checked and it seems that there is no description of EdU at line 360. The sentence you suggested to modify should be at line 350. We have adopted your suggestion and supplemented the description of this result in more detail (lines 350 - 354); We have corrected all the places where "glucose and lipid metabolism" appeared in the full text, such as lines 96, 440, 442, 453 and 680.